# Dermal Exposure to Hazardous Chemicals in Baby Diapers: A Re-Evaluation of the Quantitative Health Risk Assessment Conducted by The French Agency for Food, Environmental and Occupational Health and Safety (ANSES)

**DOI:** 10.3390/ijerph19074159

**Published:** 2022-03-31

**Authors:** Alfred Bernard

**Affiliations:** Institut de Recherche Expérimentale et Clinique (IREC), Université catholique de Louvain, 1200 Brussels, Belgium; alfred.bernard@uclouvain.be; Tel.: +32-476-476849

**Keywords:** diaper, dioxin, dioxin-like polychlorobiphenyls, polycyclic aromatic hydrocarbons, formaldehyde, fragrance

## Abstract

In January 2019, the French Agency for Food, Environmental and Occupational Health and Safety (ANSES) published an opinion on risks related to the presence of hazardous chemicals in infant diapers. ANSES found that health reference values were largely exceeded for polycyclic aromatic hydrocarbons (PAHs), dioxins (PCCD/Fs) and dioxin-like polychlorobiphenyls (DL-PCBs). The levels of formaldehyde and some fragrances were also considered potentially unsafe. Therefore, ANSES concluded that actions have to be taken to restrict levels of these contaminants in diapers. Under the exposure scenario deemed the most reliable by ANSES, estimates of cancer risks of the most potent PAHs detected in diapers exceeded 10^−3^ and hazard quotients for neurobehavioral effects attained values up to 66. Regarding dioxins and DL-PCBs, ANSES derived a hazard quotient of 12 for the risk of decreased sperm count at adult age. The aim of this study was to examine whether the exposure and risk assessment conducted by ANSES contained potential flaws that could explain such a high exceedance of health reference values. This study also put into perspective the exposure from diapers with that from breast milk whose benefits for children’s health are undisputable despite contamination by PAHs, dioxins and DL-PCB_S_.

## 1. Introduction

Disposable diapers have improved the quality of life of babies and of their caregivers so much that today having access to diapers has become a basic need. Diapers are made of several layers of materials with different functional properties. The core of diapers contains superabsorbent materials that absorb and retain the urine, keeping the skin dry and clean. Modern diapers offer health benefits by reducing the risks of diaper dermatitis, which is one of the most common skin diseases during infancy [1,2]. The use of diapers also reduces the risks of skin infection and enteric pathogen contamination of hands and the environment [2]. Over the last two decades, there have been significant innovations in the manufacturing of baby diapers. Nowadays, diapers are much thinner and much more absorbent than they were in the past. These improvements are due to the reduction of wood pulp and the use of sodium polyacrylate, a strong synthetic absorbent that limits the amount of liquid that can migrate from the diaper to the skin of the baby, resulting in a skin rewetting fraction (rewet factor) mostly below 1% [3,4]. Another significant change concerns the bleaching of wood pulp, which no more uses elemental chlorine, a possible source of contamination by dioxins [5]. Today, bleaching uses elemental chlorine-free or total chlorine-free methods that prevent the formation of dioxins or dioxin-like compounds [6].

Recently, the safety of disposable baby diapers has been questioned by a report of the French Agency for Food, Environmental and Occupational Health and Safety (ANSES). On the basis of the chemical concentrations reported by two French laboratories, ANSES conducted a quantitative health risk assessment of various contaminants detected or quantified in disposable diapers [7]. ANSES found that health reference values (HRVs) were exceeded for dioxins (PCCD/Fs), dioxin-like polychlorobiphenyls (DL-PCBs) and polycyclic aromatic hydrocarbons (PAHs). The French agency concluded that long-term health risks cannot be excluded for babies and that regulatory actions have to be taken to ensure the safety of diapers. Therefore, in October 2020, ANSES submitted to the European Chemical Agency (ECHA) a dossier for restricting levels of these hazardous substances in diapers [8]. Of note, excess cancer risk estimates calculated by ANSES were several orders of magnitude above the recommended limits while hazard quotients (HQ) attained values higher than 50. Such estimates, if proved to be correct, would be of concern as disposable diapers have been used for decades by almost all children in wealthy countries. This paper critically reviews the ANSES risk assessment by examining the accuracy of exposure data and the different assumptions made in risk calculations. The likelihood of health risks will also be assessed by comparing the exposure from diapers with that from breast milk, an important source of dioxins, PAHs and other lipophilic contaminants for breastfed babies.

## 2. Materials and Methods

The risk assessment conducted by ANSES was based on chemical analyses of baby disposable diapers that were performed in 2016, 2017 and 2018 by the French National Institute of Consumption and the Joint Laboratory Service [7]. These laboratories analyzed a variety of chemical substances by extracting them with an organic solvent or synthetic urine from whole diapers, shredded diapers or shredded parts of diapers (e.g., elastics and sheets). On the basis of these extractions, ANSES adopted three different exposure scenarios and calculated the daily intakes of substances by the dermal route with the following equations:

Scenario 1, extraction with an organic solvent from shredded diapers or shredded parts of diapers:
(1)$$\mathrm{DI}\;=\;\frac{C\;\times \;W\;\times \;N\;\times \;T\;\times \;A}{\mathrm{Bw}}$$
scenario 2.1, extraction with synthetic urine from shredded diapers:(2)$$\mathrm{DI}\;=\;\frac{C\;\times \;W\;\times \;N\;\times \;R\;\times \;A}{\mathrm{Bw}}$$ scenario 2.2, extraction with synthetic urine from whole diapers:
(3)$$\mathrm{DI}\;=\;\frac{C\;\times \;W\;\times \;N\;\times \;A}{\mathrm{Bw}}$$
where DI is the daily intake (pg, μg or mg/kg bw/day); C is the concentration of a chemical in the diaper (pg, μg or mg/kg); W is the weight of the diaper (kg); N is the number of diapers used per day; T is the fraction transferred to the skin (%); R is the rewet factor (%); A is the fraction absorbed by the skin (%); Bw is the body weight of the baby (kg).

The extraction of chemical substances from shredded diapers was performed by sinking 1 g of a shredded diaper in 100 mL of synthetic urine prepared as described by Colón et al. [9] at 37 °C for 4 h. For the extraction from whole diapers, the diapers were soaked three times with 200 mL of synthetic urine at 15 min intervals. The diapers were then placed in an oven at 37 °C for 16 h. Between 220 and 250 mL of synthetic urine were recovered. A total of 19 diapers from different brands were tested according to these procedures. The procedures for the extraction with an organic solvent were not described in the ANSES report for confidentiality reasons [7].

Similarly to a previous study on diapers and tampons [5], ANSES first performed a screening level estimate by applying the worst-case scenario to all substances quantified or detected in diapers. Then, for the substances exceeding the levels of concern, a refined assessment was conducted for every 6 months period of age of the infant by incorporating more realistic assumptions for skin transfer under scenario 1 (T value of 7% instead of 100%) and for the rewet factor under scenario 2.1 (R value of 1.32% instead of 100%). This refined analysis showed that, as expected, health risks were highest in youngest infants aged 0–6 months. This study, therefore, focused on this period of age for which the exposure parameters selected by ANSES were body weight of 3.9 kg, diaper weight of 0.024 kg and the use of 7.98 diapers per day.

ANSES based its risk assessment on scenario 2.2 deemed the most reliable by the French agency. The International Disposables and Nonwovens Association (EDANA) considers, however, that a realistic scenario should include an appropriate rewet factor [6]. For the purpose of comparison, this study assessed health risks of diapers by using the equation of scenario 2.1 with a rewet factor of 1% with regard to all the substances detected or quantified by ANSES in the organic solvent or synthetic urine extracts of shredded or whole diapers. This rewet factor of 1% is conservative since in the study of Dey et al. [4] the proportion of urine resurfacing back to the top sheet under pressure averaged 0.46% with a range of 0.32–0.66%. As performed by ANSES, skin absorption of 100% was assumed for lipophilic substances such as PAHs, dioxins and DL-PCBs. For chemicals known to be poorly absorbed by the skin, fractional absorption data derived in vivo or in vitro for human skin or non-human primate skin was used.

For substances with a threshold effect, ANSES estimated health risks by calculating the hazard quotient, which is the ratio of the daily intake to the tolerable daily intake as follows:
(4)$$\mathrm{HQ}\;=\;\frac{D}{\mathrm{TDI}}$$ where HQ is the hazard quotient; DI is the daily intake and TDI is the tolerable daily intake, both expressed in pg, μg or mg/kg body weight.

For substances assumed to have no threshold effects (PAHs), ANSES estimated the excess cancer risk using the following equation:(5)$$\mathrm{ECR}\;=\;\frac{\mathrm{DI}\;\times \;\mathrm{CSF}\;\times \;T\;\times \;\mathrm{ADAF}}{70}$$ where ECR is the excess cancer risk; DI is the daily intake (μg/kg bw/day); CSF is the cancer slope factor (μg/kg/day)^−1^; T is the duration of exposure (years); 70 is the duration of lifetime conventionally set at 70 years; ADAF is the age adjustment factor set at default values of 10 for age group < 2 years and 3 for age group 2–<16 years) [10].

This study also calculated the excess skin cancer risk of PAHs using a dermal CSF of 3.5 (μg/cm^2^/day)^−1^ [11] and assuming the skin surface area in contact with the diaper of 234 cm^2^ [12,13]. In order to consolidate the cancer risk assessment of PAHs, this paper also calculated the margin of exposure (MOE) of PAHs from diapers using the same approach as that followed by the EFSA for food products [14]. Table 1 compares the risk assessment methodology used by ANSES with that adopted in this re-evaluation.

The largest number of detected or quantified substances was found in organic solvent extracts of shredded diapers, including volatile organic compounds (naphthalene, styrene, toluene, 1,2,3-chlorobenzenes, p-isopropyl toluene, xylenes, chlorobenzene), pesticides (hexachlorobenzene, quintozene and its metabolite pentachloroaniline, glyphosate and its metabolite AMPA), formaldehyde, PCDD/Fs, DL-PCBs and perfumes (benzyl alcohol, benzyl salicylate, coumarin, hydroxyisohexyl 3-cyclohexene carboxaldehyde (Lyral^®^), butylphenyl methylpropional (Lilial^®^), limonene, linalol, alpha-isomethyl ionone). In organic solvent extracts of shredded parts of diapers, only PCDD/Fs and PAHs were found. The substances detected or quantified in simulated urine extracts of shredded or whole diapers were PCDD/Fs, DL-PCBs, PAHs and formaldehyde.

This study concentrated on PAHs, PCCD/Fs and DL-PCBs for which HRVs in the ANSES report (HQ of 1 and ECR of 10^−6^) were exceeded in all age groups of children. This re-evaluation also considered substances for which HRVs were exceeded only during the first year of life or for which there was a risk of HRVs exceedance when aggregating intake from diapers with that from other potential sources of exposure. In the ANSES report, this included hexachlorobenzene, 1,2,3-trichlorobenzene, 1,2,4-trichlorobenzene, formaldehyde, hexachlorobenzene, 1,2,4-trichlorobenzene and all the detected fragrances.

## 3. Dioxins and Dioxin-like PCBs

### 3.1. Concentrations in Diapers

Three PCDD congeners and six PCDF congeners were found in diapers in detectable or quantifiable concentrations. The two most potent dioxins, TCDD and 1,2,3,7,8-pentachlorodibenzo-p-dioxin, were not detected. Almost all DL-PCB congeners were detected or quantified. Table 2 shows the TEQ concentrations of PCDD/Fs, DL-PCBs and of the sum of PCDD/Fs and DL-PCBs (total TEQ) in diapers under the different extraction scenarios. There are several intriguing observations in these results that deserve further investigation. First, the TEQ concentrations are, if not similar, much greater (under scenario 2.1) than those reported 30 years ago when diapers were suspected of being bleached with elemental chlorine and when background environmental levels of PCDD/Fs and DL-PCBs were much higher [5]. Surprisingly, TEQ concentrations did not differ between the extracts with an organic solvent and those with synthetic urine despite the fact that PCDD/Fs and PCBs are notoriously insoluble in water. By contrast, the patterns of congeners were sharply different between the two modes of extraction. In an organic solvent extract of diapers, PCCD/Fs and DL-PCBs almost equally contributed to the total TEQ, which is frequently the case in environmental or biological matrices [15]. Rather, in the extracts with synthetic urine, DL-PCBs accounted for almost 90% of the total TEQ in whole diapers whereas they contributed to only 10% of the total TEQ in shredded diapers. It is unclear why the proportions of PCDD/Fs and DL-PCBs are inverted when extracting them under the same conditions from shredded or whole diapers. These illogical and atypical findings unavoidably raise questions about the accuracy of PCCD/Fs and DL-PCBs exposure data used by ANSES. Contrary to the common practice, TEQ concentrations of PCDDs and PCDFs were not reported separately, but from the list of detected or quantified congeners provided in the ANSES report [7], it can be deduced that PCDFs accounted for more than 50% of the PCDD/Fs TEQ. ANSES, however, provided separately the concentrations of PCB 126, which accounted for almost 90% of the DL-PCBs TEQ.

### 3.2. Toxicokinetics

Because of their lipophilicity, dioxins and DL-PCBs are usually well-absorbed by all routes of exposure [18,19]. The fractional oral absorption of dioxins and DL-PCBs varies between approximately 50% and 100% depending on the ingested dose, the duration of exposure and the degree of chlorination, the lower chlorinated congeners being better absorbed than the higher chlorinated ones. In vivo studies in animals indicate similarly high fractional dermal absorption in the range of 40–60% depending on the dose and duration of exposure. In vitro studies with human skin suggest, however, that the dermal absorption would be less important, especially in matrices such as soil or textiles [18,19]. Because wood pulp used in the absorbent core of diapers is a mixture of organic fibers, it is likely that dioxins are strongly bound to these fibers and therefore are not readily absorbed. For this reason, De Vito and Schecter [5] assumed a fractional dermal absorption of 3% for dioxins from diapers, a value that they considered conservative. Once absorbed, dioxins are readily distributed to all organs and over time they accumulate in liver and adipose tissue. The metabolism of dioxins and DL-PCBs is extremely slow, the higher chlorinated congeners being particular resistant to xenobiotics-metabolizing enzymes. The main routes of excretion are via the bile and feces. Human milk is an additional important route of excretion since the TEQ concentrations of PCDD/Fs and DL-PCBs are approximately the same in the lipid fraction of all tissues and body fluids, including milk. During lactation, the body burden of the mother decreases as a result of dioxin transfer to the nursed child [20]. The half-life of dioxins greatly varies with the degree of chlorination but also with the age, ranging from less than one year in infants up to 30 years in old adults [18,19,21].

### 3.3. Critical Adverse Effects

Two basic concepts underlie the risk assessment of dioxins and DL-PCBs. The first concept is that all toxic effects of dioxins and DL-PCBs are mediated by the sustained activation of the intracellular aryl hydrocarbon receptor (AhR). The effects of individual congeners are assumed to be additive after adjustment for their potency to activate the AhR. This adjustment is made using toxic equivalency factors (TEF) established relatively to TCDD, the most potent congener. Recent studies, however, have revealed large between-species differences in the potency of dioxins and PCB congeners to activate the AhR. For instance, while the relative effective potencies (REP) of PCBs 126, 118 and 156 derived from rat lung cells are in good accordance with the WHO_2005_ TEF, tested on human lung cells, PCB 126 elicited a 10–100-fold lower AhR-mediated activity while PCB 118 and PCB 156 were almost inactive [22]. There is thus clearly a need to develop human-specific REP/TEF based on toxicologically relevant endpoints [23,24]. The second important concept in dioxins and DL-PCBs toxicology is the body burden concept assuming that whatever the duration of exposure and the adverse outcome, risks are determined by the amount of dioxins and DL-PCBs accumulated in the body over time. In case of acute exposure, as in the Seveso accident, there is nevertheless some uncertainty regarding the respective influence of the peak exposure and of the chronic exposure in the years following the accident [25].

Dioxins and DL-PCBs are non-genotoxic carcinogens which above a certain threshold promote cancer development at various sites including the skin, the ovaries and the liver. TCDD, 2,3,4,7,8-PCDF, PCB 126 and DL-PCBs have been classified by the IARC as human carcinogens [26]. Dioxins and DL-PCBs also exert a wide range of non-carcinogenic effects including cutaneous, hepatic, neurological, immunological, reproductive, endocrine and developmental effects. Of these, the effects on male fertility and on the thyroid function during childhood exposure are regarded as the most critical.

Recently, the EFSA derived a tolerable weekly intake (TWI) of 2 pg TEQ/kg bw on the basis of the chronic Russian Children’s Study, showing that peripubertal serum TCDD and PCDDs TEQ were associated with lower sperm concentration, total sperm count and total motile sperm count measured 10 years later in healthy young men [15,27]. It should be noted that in this Russian study, there were no independent associations between semen parameters and serum levels of the PCDFs TEQ, the DL-PCBs TEQ or the total TEQ. According to the authors, this suggests that the association of dioxins with decreased male fertility might be specific to the PCDDs TEQ [27]. According to the EFSA’s expert panel, the lack of associations with the PCDFs TEQ and the DL-PCBs TEQ in the Russian study might be explained by the much lower AhR-activating potency of some congeners of dioxins and DL-PCBs [15]. Recently, indeed, Strapácová et al. [22] found that human lung REP for 2,3,4,7,8-pentachlorodibenzofuran and PCB 126 were 10–100 times lower than the respective rat lung REP on the basis of the current WHO_2005_ TEF values. This inevitably has implications for risk assessment as PCB 126 is a DL-PCB congener contributing the most to the DL-PCBs TEQ. PCB 126 also contributes to about one third of the total TEQ activity in human biological samples, including the serum of boys in the Russian study. The EPA derived a slightly higher oral reference dose (RfD) of 0.7 pg/kg/day for TCDD on the basis of the Seveso study by Mocarelli et al. [28] reporting decreased sperm concentrations and decreased motile sperm counts in men acutely exposed to TCDD at the time of the Seveso accident [25]. For assessing health risks of mixtures of PCCD/Fs and DL-PCBs, both the EFSA and the EPA recommend the use of TEF values.

No regulatory body has so far established HRVs for TCDD by the inhalation or the dermal route of exposure. In contrast to PAHs (see below), there is some reason to believe that dermal exposure to dioxins and DL-PCBs might cause similar systemic effects as oral exposure, which may legitimate the dermal–oral route extrapolation made in the ANSES report. A variety of systemic effects have indeed been observed in mice following repeated dermal exposures to dioxins including conjunctival inflammation, fibrosarcoma, thymus atrophy, liver fatty degeneration and bronchiolar adenomatoid changes [18,19].

It should be noted that ANSES initially based its risk assessment of dioxins and DL-PCBs on the RfD of the EPA, presumably for the purpose of consistency as ANSES also adopted the HRVs of the EPA for assessing risks of PAHs [7]. However, in its restriction proposal, ANSES decided to calculate the concentration limit of dioxins and DL-PCBs in diapers using the TDI established by the EFSA [8]. This study took into account this latest position of ANSES and recalculated the HQ values of dioxins and DL-PCBs on the basis of the TDI of the EFSA.

### 3.4. Evaluation

Table 2 shows the risk estimates made by ANSES for PCDD/Fs, DL-PCBs and the sum of PCDD/Fs and DL-PCBs (total TEQ) under the different extraction scenarios. Under scenario 2.2, the most reliable according to ANSES, the hazard quotient (HQ) for the total TEQ reached a value of 11.8, while for the two other scenarios it remained below 1. Scenario 2.2 is based on the assumption that all the PCDD/Fs and DL-PCBs in diapers reach the baby’s skin (rewet factor of 100%) and are entirely absorbed, resulting in a systemic bioavailability of 100% for the total TEQ content of diapers. Such a very conservative assumption is likely to greatly overestimate the intake as modern diapers are designed to retain the maximum amount of urine. If one adopts a more realistic assessment accounting for the rewet factor of 1% recommended by Dey et al. [4], Table 3 shows that all the HQ values under scenario 2.2 fall below 1.

However, even though the systemic bioavailability of dioxins and DL-PCBs would be 100%, HQ estimates by ANSES remain questionable in light of observations in the Russian study, in which the decreased sperm count correlated only with the PCDDs TEQ [27]. If, as suggested by the authors, the association with sperm count is specific of PCDDs, then the HQ should be calculated with the PCDDs TEQ only, which would decrease the HQ to approximately 0.7 under scenario 2.2 (PCDFs account for less than 50% of the PCDD/Fs TEQ, see the concentrations in diapers above). Alternatively, if, as suggested by the EFSA expert panel [15], the lack of correlation with DL-PCBs is due to the 10–100-fold lower potency of DL-PCBs to activate the AhR of human cells as compared to rat cells, then under scenario 2.2 the HQ values fall below 1 in Table 2 and even below 0.01 in Table 3 when incorporating the rewet factor of 1%.

The overestimation of dioxins and DL-PCBs risks in the ANSES report is supported by the comparison of the intake from diapers with that from breast milk. As shown in Table 2, dietary intakes of the total TEQ by nursed infants are 74.8, 336 and 6.1 times greater than the daily intakes from diapers under extraction scenarios 1, 2.1 and 2.2, respectively. These figures were based on the concentrations of dioxins and DL-PCBs in human milk measured in France in 2010 [16]. In order not to bias the comparison by the downwards temporal trends of dioxin exposure, these concentrations were adjusted to the year 2017 by assuming for the most recent years an annual decline of 10% [17]. Similar human milk/diaper intake ratios were found with milk samples analyzed in nine European countries in 2014–2015 [15]. These ratios ranged from 58 to 116, from 261 to 528 and from 4.8 to 9.5 under ANSES extraction scenarios 1, 2.1 and 2.2, respectively. As shown in Table 3, the total TEQ intake by nursed infants are more than 400 times greater than the intake from diapers under the three ANSES scenarios when the latter are calculated by incorporating the rewet factor of 1%. These calculations make unrealistic the assumption by ANSES that diapers contribute to 10% of the total TEQ intake [8]. It is also interesting to compare concentrations in human milk with the concentration limit proposed by ANSES in its restriction proposal [8]. ANSES recommends the maximum concentration for the total TEQ in diapers of 0.7 pg/kg diaper. This TEQ concentration is 171–342 times lower than the concentrations in human milk observed in 2014–2015 in nine European countries (range, 120–240 pg TEQ/kg) [15].

There is no evidence that at the current exposure levels in the European Union dioxins and DL-PCBs in breast milk reduce the future fertility of breastfed boys. On the contrary, in a study conducted among adolescents, breastfeeding was associated in a dose-dependent manner with an increase in serum inhibin B, a marker of fertility at adult age that was measured with two different immunoassays [29]. The increase averaged 20% in adolescents who were breastfed for more than six months. The lack of adverse effects of breastfeeding on male fertility despite the relatively high concentrations of dioxins and DL-PCBs in breast milk might be explained by the short half-lives of these contaminants during early life, which, coupled with the children’s rapid growth, prevents an excessive increase in the dioxin body burden [18,19,21]. In the Seveso cohort acutely exposed to TCDD, a decrease in sperm count was found in men who had been breastfed in the years following the accident. Mothers of the children with decreased sperm count had, however, a median serum level of TCDD of 58.9 pg/g fat, which was approximately 10 times higher than the TCDD background serum levels in the 1970s and which is more than 50 times higher than the current levels of TCDD in the serum and breast milk [28]. Concentrations of dioxins and DL-PCBs in breast milk have considerably decreased over the last fifty years, with an average annual decrease of 6% between 1972 and 2011, according to a Swedish study [17]. This temporal trend parallels the decline of sperm count in Western countries, which has been estimated at 1.4% per year between 1973 and 2011 [30]. These parallel downwards temporal trends of human dioxin exposure and sperm count argue against the hypothesis implicating dioxins as a significant driver of the global decline of the Western men’s fertility. These temporal trends also make unrealistic the risk of decreased male fertility calculated by ANSES for dioxins in diapers, which contribute to infant exposure more than 100 times less than breast milk. The possibility of nonmonotonic relationships could be invoked to causally link these parallel downwards trends but this would not be consistent with the dose-dependent decrease in sperm count with increasing dioxin exposure observed in the Russian study [27].

## 4. Polycyclic Aromatic Hydrocarbons (PAHs)

### 4.1. Concentrations in Diapers

As shown in Table 4, concentrations of PAHs in diapers in the ANSES report ranged from 249 to 598 μg/kg [7]. These concentrations actually were not quantified but corresponded to the half of the limits of quantification (LOQ) of individual PAHs. The analytical method used by French laboratories had indeed relatively poor sensitivity with limits of detection (LOD) of individual PAHs higher than 150 μg/kg. Actually, this LOD is two orders of magnitude higher than the LODs of methods recommended to measure PAHs in foodstuffs [14]. This surrogate method for evaluating exposure explains the atypical concentration pattern of PAHs in diapers showing little variation between individual PAHs while in most environmental or biological matrices individual PAH concentrations differ by a factor of 10–100. Under these conditions, it is puzzling that ANSES recommends a concentration limit as low as 2.7 ng TEQ/kg diaper for the sum of PAHs [8]. In the case of benzo[a]pyrene and dibenz[a,h]anthracene (TEF of 1), this concentration limit is five orders of magnitude lower than the LOD of the analytical method used in the ANSES risk assessment. This concentration limit for diapers is even 10 times lower than the median LOD of the methods used to measure PAHs in food products [14].

No study could be identified in the peer-reviewed literature to compare the PAH concentrations in diapers used by ANSES with those from other sources. In 2019, however, the Federal Food Safety and Veterinary Office in Switzerland conducted a survey of PAHs in diapers using, after some modifications, an analytical method that can correctly quantify PAHs in food samples [33]. The concentrations of PAHs measured in the absorbent core of five brands of diapers ranged from < 0.15 to 4.6 μg/kg and the concentrations of the two most potent PAHs, benzo[a]pyrene and dibenz[a,h]anthracene, ranged from < 0.12 to 1.4 μg/kg. Similarly low concentrations were found in the non-absorbent parts of the diapers, the most potent PAHs even being undetectable in most cases. The values reported in the Swiss report are thus 2–3 orders of magnitude lower than those at the basis of the quantitative health risk assessment performed by ANSES.

### 4.2. Toxicokinetics

PAHs are lipophilic substances that are usually well-absorbed by all routes [10,34]. The dermal absorption varies depending on the species, the anatomic site, the solvent or the vehicle and the type of the experimental study (in vivo or ex vivo). Human studies testing PAHs in an organic solvent have reported fraction dermal absorption up to 80% but it is likely that the skin absorption of PAHs is enhanced by the use of an organic solvent. The fractional dermal absorption of PAHs can reasonably be assumed to be in the range of 10–60%. Absorbed PAHs are distributed in the whole body and especially in lipid-rich organs. In the skin, like in internal organs, PAHs are rapidly metabolized by cytochrome P-450-dependent enzymes into water-soluble compounds, which can be further transformed into conjugates. The metabolism of PAHs involves the formation of electrophilic intermediates that can bind to DNA and initiate tumors. PAHs metabolites and their conjugates are eliminated via the urine and feces with short half-lives. Similarly to dioxins, breastfeeding is an additional route of excretion [10,34].

### 4.3. Critical Adverse Effects

#### 4.3.1. Carcinogenic Effects

The data on the carcinogenicity of PAHs in humans essentially derive from studies among workers with high inhalation or dermal exposure to mixtures of PAHs. There are no adequate carcinogenicity data for human exposure by ingestion. In contrast, there is ample evidence of increased risks of lung cancer in occupations involving exposure to PAHs mixtures containing benzo[a]pyrene such as aluminum production, chimney sweeping, coal gasification, coal tar distillation, coke production, iron and steel founding, paving and roofing with coal tar pitch. All these occupational exposures are classified as carcinogenic to humans. In the industry, an increased risk of skin cancer (including the scrotum) has been documented following high dermal exposure to mixtures of PAHs including benzo[a]pyrene, such as soot, coal tar, shale oils, coal tar pitches and unrefined mineral oils [35].

In animals, like in humans, the sites of tumors induced by PAHs are largely determined by the route of exposure. By inhalation or intratracheal administration, benzo[a]pyrene induces only respiratory tract cancers. Upper digestive tract tumors were observed in some inhalation studies, but this is most probably the consequence of the mucociliary clearance of inhaled PAHs. In chronic oral bioassays, benzo[a]pyrene induces mainly tumors of the digestive tract (forestomach, esophagus, tongue, larynx, liver). By the dermal route, lifetime carcinogenicity bioassays in several strains of mice have demonstrated that benzo[a]pyrene induces only skin tumors. These studies involved two or three times/week exposure protocols, at least two exposure levels plus controls and histopathological examinations of the skin and internal organs (for review see [10,34]). As there is no experimental evidence suggesting dermally applied PAHs may increase the risk of systemic tumors, skin cancer should be considered as the critical effect for the dermal exposure to PAHs. The lack of systemic carcinogenicity of PAHs applied to the skin, even at very high doses, can be explained by the rapid metabolism of PAHs in the skin. A recent study using a realistic human ex vivo skin model has shown that less than 3% of benzo[a]pyrene applied to the skin is bioavailable in the unmetabolized form, the only form susceptible to initiate cancer systemically [36]. This percentage might still be over-estimated as one can expect that human skin has a lower metabolic activity ex vivo than in vivo. Benzo[a]pyrene is a complete carcinogen acting as both an initiator and a promoter of carcinogenesis. The mechanism by which benzo[a]pyrene induces carcinogenicity is through its mutagenicity, a mechanism that presumably applies to all types of tumors, regardless of the route of exposure [10].

The EPA established a cancer slope factor (CSF) of 1 mg/kg/day^−1^ for assessing human cancer risk associated with lifetime oral exposure to benzo[a]pyrene [10]. This CSF was derived from the digestive tract tumors observed in mice. In addition, for the assessment of cancer risks during early life, the EPA recommends the application of an age-dependent adjustment factor (ADAF) to account for the fact that benzo[a]pyrene is a genotoxic carcinogen and also that the CSF is based on carcinogenic effects observed in adult animals.

#### 4.3.2. Developmental Effects

Animal studies demonstrate that exposure to benzo[a]pyrene is associated with developmental (including neurotoxic), reproductive and immunological effects. In addition, epidemiological studies involving exposure to PAHs mixtures have reported associations between biomarkers of benzo[a]pyrene exposure (benzo[a]pyrene diol epoxide-DNA adducts) and adverse birth outcomes, neurobehavioral effects and decreased fertility. These adverse effects in humans and animals have been demonstrated by inhalation or oral exposure to benzo[a]pyrene. The EPA established an oral RfD of benzo[a]pyrene of 0.3 μg/kg/day on the basis of neurobehavioral changes in rats exposed to benzo[a]pyrene during early life [10].

### 4.4. Evaluation

PAHs are a family of toxicants that are ubiquitous in the environment. These contaminants are of concern because several PAH congeners are highly carcinogenic in laboratory animals and have long been recognized as potent human carcinogens, causing lung cancer by inhalation and skin cancer by dermal exposure. Table 4 based on the data from the ANSES report [7] shows that HQs for the developmental effects of the most potent PAHs (benzo[a]pyrene and dibenzo[a,h]anthracene) detected in diapers reach values higher than 50 while cancer risk estimates reach values exceeding 10^−3^. Even when adjusting the intake from diapers for the rewet factor of 1% (Table 5), levels of cancer risks remain totally unacceptable, especially when cumulated over three years of diaper use. These estimates rely on the assumption that dermally absorbed PAHs can cause systemic cancers as observed in animals with chronic oral exposure. This assumption, however, is strongly challenged by animal studies showing that PAHs applied dermally induce only skin tumors. Thus, cancer risk estimates should be calculated not with the oral but with the dermal CSF. The estimates of skin cancer risks of PAHs calculated with the dermal CSF of 3.5 μg/cm^2^/day^−1^ proposed by Knafa et al. [11] are shown in Table 6. Even when incorporating the rewet factor of 1%, these estimates are even higher than those based on the oral CSF.

It is, of course, inconceivable that diapers could be contaminated by PAHs at levels entailing cancer risks comparable to those observed in the industry, even in the industries not complying with the maximal risk level for occupational exposure set by the European Commission (10^−4^) [37]. These unrealistic estimates cannot be explained by the TEF approach adopted by ANSES that the EFSA considers scientifically invalid because of lack of data for the oral carcinogenicity of individual PAHs [14]. As shown in Table 7, the alternative margin of exposure (MOE) approach used by the EFSA for assessing risks of PAHs in food products also leads to totally unacceptable risks. The MOE calculated with the EDANA equation (rewet factor of 1%) for the intake of benzo[a]pyrene (MOE, 342) and of the 8 PAHs carcinogenic in animals (MOE, 471) are largely below the acceptable levels (MOE of at least 10,000, according to the EFSA [14]). Furthermore, alone, the overly conservative assumptions made in the ANSES report cannot explain risks of that magnitude. The only possible explanation lies in the combined use of the overconservative scenario 2.2 and of the LOQ/2 values of an inadequate analytical method as surrogate concentrations in diapers. The survey conducted in 2019 by the Swiss Federal Food Safety and Veterinary Office suggests that values of PAHs in diapers used by ANSES have been overestimated by 2–3 orders of magnitude [33].

Two lines of evidence add support to the view that the ANSES report unreasonably overestimated cancer risks of PAHs in diapers. The first comes from the comparison of PAHs intake from diapers with that from human milk, a comparison made possible if one assumes, as does ANSES, that PAHs cause systemic cancers irrespective of the route of exposure. When incorporating the rewet factor of 1%, the intake of PAHs from diapers is rather similar to that from human milk (Table 5). However, if concentrations of PAHs were overestimated by more than two orders of magnitude as suggested by data from Switzerland [33], the intake from mother milk should be more than 100 times greater than that from diapers. There is no epidemiological evidence whatsoever associating breastfeeding with increased risks of cancer or developmental effects. On the contrary, breastfeeding is unanimously recognized as protective against various diseases including cancers (leukemia) and as beneficial to the child’s neurodevelopment, improving the IQ and reducing the risk of behavioral disorders [38,39].

The second line of evidence comes from epidemiological or case report studies that provide no indication at all of adverse effects of diapers despite decades of use by almost all children in wealthy countries. If diapers were to pose cancer risks as high as 10^−3^ as suggested by the ANSES report, it is hard to believe that such risks could have passed undetected after such a long and widespread use as the large-scale use of disposable diapers started in the USA in 1961 [40]. In the anogenital region, the highly permeable scrotum has long been known to be particularly sensitive to the carcinogenic effects of PAHs. Squamous cell carcinoma (SCC) is the type of scrotal malignancy caused by occupational exposure specifically to PAHs. With preventive measures implemented at workplace, SCC has become a very rare cancer with a steady incidence rate through the 20th century. The main non-occupational risk factors of SCC are sun exposure, human papilloma virus and several types of treatment for skin diseases. As the median range at diagnosis is 52–57 years, it appears unlikely that SCC could be initiated during infancy, even if the median SCC latency is close to 30 years [41].

## 5. Other Compounds Present in Diapers at Potentially Unsafe Levels According to ANSES

Table 8 presents the ANSES risk assessment for substances that exceeded HRVs during the first year of life only (HICC, hydroxyisohexyl cyclohexene carboxaldehyde; BPMP, butylphenyl methylpropional) or for which there was a risk of HRVs exceedance when aggregating exposure from diapers with that from other sources. In the ANSES report, this also included the hexachlorobenzene pesticide and 1,2,4-trichlorobenzene, but the presence of these compounds in the list was due to calculation errors by ANSES (the daily intakes per kg body weight instead of decreasing were increasing with child’s age). Except for formaldehyde, these substances were detected in extraction scenario 1 (with an organic solvent) and not found in scenarios 2.1 and 2.2. The risk assessment in Table 8 is flawed due to two major failures. First, similarly to PAHs, fragrance chemicals were not measured with an adequate analytical method and therefore, again, ANSES based its risk assessment on LOQ values, which are a poor proxy of the effective exposure. Second, ANSES experts seem to have overlooked that the stratum corneum of the skin is primarily a physiological barrier that only lipophilic and uncharged molecules can easily cross. Obviously, several of the substances listed in Table 7 are too poorly absorbed across skin to reasonably assume a fractional absorption of 100%. This is particularly true for formaldehyde which is too reactive and too rapidly metabolized in the skin to be available systematically, especially at the very low concentrations found in diapers [42,43,44]. Applied dermally, formaldehyde reacts instantaneously with skin constituents to form a variety of derivatives including adducts and cross-links. This local reactivity is also reflected by the skin’s sensitizing properties of formaldehyde and the development of contact dermatitis in humans. Formaldehyde is also rapidly metabolized in the skin, which further reduces the systemic availability of unreacted formaldehyde. However, even assuming that a small fraction of formaldehyde in diapers is dermally absorbed, the amount distributed to the body would be totally insignificant compared to that produced endogenously. Formaldehyde is, indeed, an intermediary metabolite essential to all cells with an endogenous daily turnover in humans estimated between 878–1.310 mg/kg bw [45]. Among the fragrances, D-limonene and benzyl salicylate are also poorly absorbed by the human skin while HICC and BPMP penetrate only to a limited extent [46,47,48,49,50].

For a meaningful assessment of the systemic effects of substances in Table 8, it is thus a prerequisite to adjust the daily intake for the dermal absorption while adjusting also HRVs for the oral absorption to properly compare the systemic doses. As shown in Table 9, these adjustments for the effective absorption result in HQ values largely below 0.1 for limonene and formaldehyde and in an MOE for benzyl salicylate higher than 10,000. Values of HQ and of the MOE_ref_/MOE for other compounds in Table 8 are between 0.1 and 1, which, according to ANSES, might not be sufficiently protective. Because these fragrances are added voluntarily and can be easily removed, ANSES decided not to include them in its restriction proposal. Only formaldehyde was included, with the concentration limit of 0.21 mg/kg diaper [8]. It should be noticed that intakes from diapers of compounds in Table 8 and Table 9 were calculated with the scenario 1 equation incorporating a skin transfer factor of 7% (except for formaldehyde in the synthetic urine extract). If one replaces this transfer factor by the rewet factor of 1% recommended by Dey et al. [4], HQ values are further reduced and the MOE is further increased by a factor of 7, which makes systemic effects unlikely for all chemicals in Table 8 and Table 9.

## 6. Conclusions

The quantitative health risk assessment conducted by ANSES is flawed by several scientifically unjustified risk assessment approaches and assumptions that have led to incredible risk estimates and to concentration limits in diapers hardly quantifiable by the current analytical methods. Even if a very conservative approach is understandable given the vulnerability of infants, it is not reasonable to assume a fractional dermal absorption of 100% and calculate systemic risks for substances like formaldehyde, which are very poorly absorbed by the skin. Similarly, assuming that all the substance present in diapers can enter in contact with the skin is not realistic in regard to the very low fraction of absorbed fluid rewetting the skin. ANSES based its risk assessment of PAHs on a route and species extrapolation, and by doing so the French agency fails to consider that dermally applied PAHs induce only skin tumors. The risk overestimation for PAHs is most likely due to the combined use of an overconservative exposure scenario and of the LOQ values of an inadequate analytical method as surrogate concentrations in diapers. There are also some doubts regarding the accuracy of dioxin and DL-PCB concentrations that display illogical and atypical patterns of congeners. Under the scenario deemed most reliable by ANSES, the total TEQ activity in diapers was, indeed, predominantly contributed by PCB 126, a congener with questionable potency and uncertain association with decreased sperm count. There is thus a clear need to revisit the ANSES risk assessment by using more accurate exposure data, more toxicologically relevant endpoints and more realistic exposure scenarios. The revisited risk assessment should also evaluate the plausibility and likelihood of adverse effects of diapers by comparing the intake from diapers with that from breast milk, which offers numerous benefits to children’s health despite much higher concentrations of dioxins, DL-PCBs and PAHs than those found in diapers.

## Figures and Tables

**Table 1 ijerph-19-04159-t001:** Comparison of the risk assessment methodology used by ANSES with that recommended in this study.

Risk Assessment Steps	Compounds	ANSES	This Study
Exposure data	All the substances quantified or detected by ANSES	Concentrations in diapers, frequency of use and weight of diapers for an infant aged 0–6 months as reported by ANSES [7]	
Exposure scenarios	PCCD/Fs and DL-PCBs	Scenario 1 (T, 7%; A, 100%)	Scenario 2.1 (R, 1%; A, 100%)
		Scenario 2.1 (R, 1.32%; A, 100%)	
		Scenario 2.2 (R, 100%; A, 100%)	
	PAHs	Scenario 2.2 (R, 100%; A, 100%)	
	Other compounds	Scenario 1 (T, 7%; A, 100%); also, scenario 2.2 (R, 100%; A, 100%) for formaldehyde	Scenario 1 (T, 7% or R, 1%) with A values derived from experimental data
Critical effects	PCDD/Fs and DL-PCBs	Reduced sperm count at adult age in humans	
		Digestive tract tumors in mice	
	PAHs	Neurobehavioral changes in rats exposed during early life	
			Skin cancers in rats
	Other compounds	Systemic effects in animals (hepatotoxicity, nephrotoxicity…)	
Risk evaluation	PCDD/Fs and DL-PCBs	Calculation of HQ with the EFSA TDI (0.3 pg TEQ/kg/d)	
	PAHs	Calculation of HQ with the EPA oral RfD (0.3 μg/kg/d)	
		Calculation of excess digestive tract cancers with the EPA oral CSF of 1 (mg/kg/d^−1^)	
			Calculation of excess skin cancer with the skin CSF of 3.5 (μg/cm^2^/d^−1^)
	Other compounds	Calculation of MOE or HQ with HRVs (NOAEL or TDI) used by ANSES	

For abbreviations, see Section 2 Materials and Methods. Critical effects are explained in Section 3.3 and Section 4.3.

**Table 2 ijerph-19-04159-t002:** Risk assessment of dioxins (PCDD/Fs) and DL-PCBs in diapers conducted by ANSES and comparison of the intake from diapers with that from breast milk.

ANSES Extraction Scenario	Compound	Concentration in Diapers (pg TEQ/kg) ^1^	Intake from Diapers (pg TEQ/kg/d) ^2^	EFSA TDI (pg TEQ/kg/d)	Hazard Quotient	Intake from Breast Milk (pg TEQ/kg/d) ^3^	Breast Milk/Diaper inTake Ratio
**Scenario 1**	PCDD/Fs	39.8	0.14		0.47	12.2	87.1
Organic solvent	DL-PCBs	43.4	0.15	0.3	0.49	9.55	63.7
Shredded diapers	PCDD/Fs + DL-PCBs	83.2	0.29		0.96	21.7	74.8
** Scenario 2.1 **	PCDD/Fs	92.0	0.0596		0.20	12.2	205
Synthetic urine	DL-PCBs	7.55	0.0049	0.3	0.02	9.55	1.953
Shredded diapers	PCDD/Fs + DL-PCBs	99.6	0.0645		0.22	21.7	336
** Scenario 2.2 **	PCDD/Fs	8.84	0.43		1.45	12.2	28.4
Synthetic urine	DL-PCBs	63.6	3.12	0.3	10.4	9.55	3.06
Whole diapers	PCDD/Fs + DL-PCBs	72.4	3.55		11.9	21.8	6.14

^1^ TEQ concentrations based on the WHO_2005_ TEF values. ^2^ Intake from diapers was calculated for an infant aged 0–6 months (body weight, 3.9 kg; 7.98 diapers/day; diaper weight, 24 g). ^3^ Intake from breast milk was based on the data of Focant et al. [16] that were adapted to year 2017 on the basis of an annual decline of 10% [17]. The PCDD/Fs and DL-PCBs daily intake was estimated for an infant of 5 kg of body weight fed daily with 700 mL of breast milk containing 25 g/L of lipids.

**Table 3 ijerph-19-04159-t003:** Risk assessment of dioxins (PCDD/Fs) and DL-PCBs in diapers with the scenario 2.1 equation incorporating a rewet factor of 1% and comparison of the intake from diapers with that from breast milk.

ANSES Extraction Scenario	Compound	Concentration in Diapers (pg TEQ/kg) ^1^	Intake from Diapers (pg TEQ/kg/d) ^2^	EFSA TDI (pg TEQ/kg/d)	Hazard Quotient	Intake from Breast Milk (pg TEQ/kg/d) ^3^	Breast Milk/Diaper Intake Ratio
** Scenario 1 **	PCDD/Fs	39.8	0.020		0.065	12.2	626
Organic solvent	DL-PCBs	43.4	0.021		0.070	9.55	448
Shredded diapers	PCDD/Fs + DL-PCBs	83.2	0.041		0.135	21.7	531
** Scenario 2.1 **	PCDD/Fs	92.0	0.045		0.152	12.2	270
Synthetic urine	DL-PCBs	7.55	0.0037	0.3	0.012	9.55	2.574
Shredded diapers	PCDD/Fs + DL-PCBs	99.6	0.049		0.163	21.7	444
** Scenario 2.2 **	PCDD/Fs	8.84	0.0043		0.014	12.2	2.811
Synthetic urine	DL-PCBs	63.6	0.031		0.105	9.55	306
Whole diapers	PCDD/Fs + DL-PCBs	72.4	0.036		0.119	21.8	612

^1^ TEQ concentrations based on the WHO_2005_ TEF values. ^2^ Intake from diapers was calculated for an infant aged 0–6 months (body weight, 3.9 kg; 7.98 diapers/day; diaper weight, 24 g). ^3^ Intake from breast milk was based on the data of Focant et al. [16] that were adapted to year 2017 on the basis of an annual decline of 10% [17]. The PCDD/Fs and DL-PCBs daily intake was estimated for an infant of 5 kg of body weight fed daily with 700 mL of breast milk containing 25 g/L of lipids.

**Table 4 ijerph-19-04159-t004:** Risk assessment of polycyclic aromatic hydrocarbons (PAHs) in diapers with the scenario 2.2 equation and comparison of the intake from diapers with that from breast milk.

HAPs	Concentration (μg/kg) ^1^	Intake from Diapers (μg/kg/d) ^2^	TEF ^3^	Intake from Diapers (μg TEQ/kg/d) ^2^	EPA Oral RfD (μg/kg/d)	Hazard Quotient	EPA Oral CSF (mg/kg/d^−1^)	Excess Cancer Risk	Intake from Breast Milk (μg/kg/d) ^4^	Breast Milk/Diaper Intake Ratio
Cyclopenta[c,d]pyrene	311	15.3	0.1	1.53		5.51		1.09 × 10^−4^		
Chrysene	249	12.2	0.01	0.12		0.41		8.76 × 10^−6^	0.12	9.8 × 10^−3^
5-methylchrysene	311	15.3	0.01	0.15		0.51		1.09 × 10^−5^		
Benzo[b]fluoranthene	381	18.7	0.1	1.87		6.24		1.34 × 10^−4^	0.11	5.9 × 10^−3^
Benzo[k]fluoranthene	369	18.1	0.1	1.81		6.03		1.29 × 10^−4^	0.88	4.9 × 10^−2^
Benzo[j]fluoranthene	369	18.1	0.1	1.81	0.3	6.03	1	1.29 × 10^−4^		
Benzo[e]pyrene	598	29.4	0.01	0.29		0.98		2.10 × 10^−5^		
Benzo[a]pyrene	405	19.9	1	19.9		66.3		1.42 × 10^−3^	0.11	5.5 × 10^−3^
Dibenzo[a,h]anthracene	311	15.3	1	15.3		51.0		1.09 × 10^−3^	0.61	4.0 × 10^−3^
Benzo[g,h,i]perylene	418	20.5	0.01	0.21		0.68		1.47 × 10^−5^	0.73	3.6 × 10^−3^
S PAHs	3.722	182		33.8		113		2.41 × 10^−3^	2.56	1.4 × 10^−2^
S 8PAHs	2.133	104							2.56	2.5 × 10^−2^
S 4PAHs	1.035	50.8							0.34	6.7 × 10^−3^

^1^ Extraction from whole diapers with synthetic urine. ^2^ Intake from diapers was calculated according to ANSES scenario 2.2 for an infant aged 0–6 months (body weight, 3.9 kg; 7.98 diapers/day; diaper weight, 24 g) by assuming a fractional dermal and oral absorption of 100%. ^3^ TEF values proposed by INERIS [31]. ^4^ Intake from breast milk was based on the data of Santonicola et al. [32] and calculated for an infant of 5 kg of body weight fed daily with 700 mL of maternal milk containing 25 g/L of lipids.

**Table 5 ijerph-19-04159-t005:** Risk assessment of polycyclic aromatic hydrocarbons (PAHs) in diapers with the scenario 2.1 equation incorporating a rewet factor of 1% and comparison of the intake from diapers with that from breast milk.

HAPs	Concentration (μg/kg) ^1^	Intake from Diapers (μg/kg/d) ^2^	TEF ^3^	Intake from Diapers (μg TEQ/kg/d) ^2^	EPA Oral RfD (μg/kg/d)	Hazard Quotient	EPA Oral CSF (mg/kg/d^−1^)	Excess Cancer Risk	Intake from Breast Milk (μg/kg/d) ^4^	Breast Milk/Diaper Intake Ratio
Cyclopenta[c,d]pyrene	311	0.15	0.1	0.015		0.05		1.09 × 10^−6^		
Chrysene	249	0.12	0.01	0.0012		0.004		8.76 × 10^−8^	0.12	1.0
5-methylchrysene	311	0.15	0.01	0.0015		0.005		1.09 × 10^−7^		
Benzo[b]fluoranthene	381	0.19	0.1	0.019		0.063		1.34 × 10^−6^	0.11	0.58
Benzo[k]fluoranthene	369	0.18	0.1	0.018	0.3	0.06	1	1.29 × 10^−6^	0.88	4.89
Benzo[j]fluoranthene	369	0.18	0.1	0.018		0.06		1.29 × 10^−6^		
Benzo[e]pyrene	598	0.29	0.01	0.0029		0.0097		2.10 × 10^−7^		
Benzo[a]pyrene	405	0.20	1	0.11		0.67		1.42 × 10^−5^	0.11	0.55
Dibenzo[a,h]anthracene	311	0.15	1	0.15		0.50		1.09 × 10^−5^	0.61	4.07
Benzo[g,h,i]perylene	418	0.21	0.01	0.0021		0.007		1.47 × 10^−7^	0.73	3.48
S PAHs	3.722	1.82		0.34		1.13		2.41 × 10^−5^	2.56	1.41
S 8PAHs	2.133	1.05							2.56	2.44
S 4PAHs	1.035	0.51							0.34	0.67

^1^ Extraction from whole diapers with synthetic urine (ANSES scenario 2.2). ^2^ Intake from diapers was calculated according to the scenario 2.1 equation incorporating a rewet factor of 1% for an infant aged 0–6 months (body weight, 3.9 kg; 7.98 diapers/day; diaper weight, 24 g; rewet factor, 1%). ^3^ ANSES used the TEF values proposed by INERIS [31]. ^4^ Intake from breast milk was based on the data of Santonicola et al. [32] and calculated for an infant of 5 kg of body weight fed daily with 700 mL of breast milk containing 25 g/L of lipids.

**Table 6 ijerph-19-04159-t006:** Risk of skin cancer from polycyclic aromatic hydrocarbons (PAHs) in diapers assessed using the scenario 2.2 equation or the scenario 2.1 equation incorporating a rewet factor of 1%.

PAHs	Concentration (μg/kg) ^1^	Intake from Diapers (μg/d) ^2^	TEF ^3^	Intake from Diapers (μg TEQ/d) ^2^	Intake from Diapers (μg/cm^2^/d) ^4^	Dermal CSF (μg/cm^2^/d^−1^) ^5^	Excess Skin Cancer Risk
Intake calculated according to the scenario 2.2 equation							
Cyclopenta[c,d]pyrene	311	59.6	0.1	5.96	2.55 × 10^−2^		6.36 × 10^−3^
Chrysene	249	47.7	0.01	0.48	2.04 × 10^−3^		5.09 × 10^−4^
5-methylchrysene	311	59.6	0.01	0.6	2.55 × 10^−3^		6.36 × 10^−4^
Benzo[b]fluoranthene	381	73.0	0.1	7.3	3.12 × 10^−2^		7.79 × 10^−3^
Benzo[k]fluoranthene	369	70.7	0.1	7.1	3.02 × 10^−2^		7.55 × 10^−3^
Benzo[j]fluoranthene	369	70.7	0.1	7.1	3.02 × 10^−2^	3.5	7.55 × 10^−3^
Benzo[e]pyrene	598	114.6	0.01	1.15	4.89 × 10^−3^		1.22 × 10^−3^
Benzo[a]pyrene	405	77.6	1	77.6	3.31 × 10^−1^		8.28 × 10^−2^
Dibenzo[a,h]anthracene	311	59.6	1	59.6	2.55 × 10^−1^		6.36 × 10^−2^
Benzo[g,h,i]perylene	418	80	0.01	0.80	3.42 × 10^−3^		8.55 × 10^−4^
S PAHs	3.722	713		168	7.16 × 10^−1^		1.79 × 10^−1^
Intake calculated according to the scenario 2.1 equation (rewet factor of 1%)							
Cyclopenta[c,d]pyrene	311	0.6	0.1	0.060	2.55 × 10^−4^		6.36 × 10^_5^
Chrysene	249	0.48	0.01	0.0048	2.04 × 10^−5^		5.09 × 10^−6^
5-methylchrysene	311	0.6	0.01	0.0060	2.55 × 10^−4^		6.36 × 10^−6^
Benzo[b]fluoranthene	381	0.73	0.1	0.073	3.12 × 10^−4^		7.79 × 10^−5^
Benzo[k]fluoranthene	369	0.71	0.1	0.071	3.02 × 10^−4^		7.55 × 10^−5^
Benzo[j]fluoranthene	369	0.71	0.1	0.071	3.02 × 10^−4^	3.5	7.55 × 10^−5^
Benzo[e]pyrene	598	1.15	0.01	0.012	4.89 × 10^−5^		1.22 × 10^−5^
Benzo[a]pyrene	405	0.78	1.0	0.78	3.32 × 10^−3^		8.28 × 10^−4^
Dibenzo[a,h]anthracene	311	0.6	1.0	0.60	2.55 × 10^−3^		6.36 × 10^−4^
Benzo[g,h,i]perylene	418	0.8	0.01	0.080	3.42 × 10^−5^		8.55 × 10^−6^
S PAHs	3.722	7.13		1.68	7.16 × 10^−3^		1.79 × 10^−3^

^1^ Extraction from whole diapers with synthetic urine (ANSES scenario 2.2). ^2^ Intake from diapers calculated for an infant aged 0–6 months (body weight, 3.9 kg; 7.98 diapers/day; diaper weight, 24 g). ^3^ ANSES used the TEF values proposed by INERIS [31]. ^4^ Intake based on the skin surface area in contact with the diaper of 234 cm^2^ [13,14]. ^5^ Dermal cancer slope factor of benzo[a]pyrene (Knafla et al. [11]).

**Table 7 ijerph-19-04159-t007:** Margin of exposure (MOE) for polycyclic aromatic hydrocarbons (PAHs) with the scenario 2.2 equation or the scenario 2.1 equation incorporating the rewet factor of 1%.

PAHs	Intake from Diapers (μg/kg/d) ^1^	BMDL10 (mg/kg/d) ^2^	MOE	MOE (1% Rewet Factor)
BaP	19.9	0.07	3.52	352
PAH4	50.8	0.34	6.69	669
PAH8	104	0.49	4.71	471

^1^ Extraction from whole diapers with synthetic urine (scenario 2.2). Intake from diapers calculated for an infant aged 0–6 months (body weight, 3.9 kg; 7.98 diapers/day; diaper weight, 24 g). ^2^ From the EFSA [14].

**Table 8 ijerph-19-04159-t008:** Risk assessment conducted by ANSES for other compounds detected or quantified in diapers at potentially unsafe levels.

Compound	Concentration (mg/kg) ^1^	Intake from Diapers (mg/kg/d) ^2^	TDI (mg/kg/d)	Hazard Quotient	NOAEL (mg/kg/d)	MOE	MOE_ref_	MOE_ref_/MOE
1,2,3 trichlorobenzene	0.25	8.59 × 10^−4^	8 × 10^−3^	0.107				
Coumarin	25	8.59 × 10^−2^	0.1	0.86				
Limonene	25	8.59 × 10^−2^	0.1	0.86				
Benzyl salicylate	25	8.59 × 10^−2^			50	582	100	0.17
HICC (Lyral^®^)	25	8.59 × 10^−2^			15	175	300	1.71
BPMP (Lilial^®^)	25	8.59 × 10^−2^			5	58.2	100	1.72
Alpha-isomethyl ionone	25	8.59 × 10^−2^			50	582	100	0.17
Formaldehyde	37.4	0.13	0.15	0.86				
Formaldehyde (synthetic urine)	2.75	0.135	0.15	0.90				

^1^ Extraction from shredded diapers with an organic solvent (scenario 1). Formaldehyde was also quantified in the extract of shredded diapers with synthetic urine (scenario 2.2). The value of 25 mg/kg for the fragrances corresponds to LOQ/2. ^2^ Intake from diapers calculated according to ANSES extraction scenario 1 (ANSES extraction scenario 2.2 was used for formaldehyde) for an infant aged 0–6 months (body weight, 3.9 kg; 7.98 diapers/day; diaper weight, 24 g). Abbreviations: HICC, hydroxyisohexyl cyclohexene carboxaldehyde; BMP, butylphenyl methylpropional; TDI, tolerable daily intake; MOE, margin of exposure; NOAEL, no-observed-adverse-effect level.

**Table 9 ijerph-19-04159-t009:** Risk assessment of other compounds detected or quantified in diapers at potentially unsafe levels according to ANSES by taking into account the fractional dermal and oral absorption.

Compound	Concentration (mg/kg) ^1^	Dermal Absorption (%) ^2^	Intake from Diapers (mg/kg/d) ^3^	TDI (mg/kg/d) ^4^	Hazard Quotient	NOAEL (mg/kg/d) ^4^	MOE	MOE_ref_	MOE_ref_/MOE
1,2,3 trichlorobenzene	0.25	100	8.59 × 10^−4^	8 × 10^−3^	0.107				
Coumarin	25	100	8.59 × 10^−2^	0.1	0.86				
Limonene	25	0.16	1.37 × 10^−4^	0.1	1.37 × 10^−3^				
Benzyl salicylate	25	0.031	0.27 × 10^−4^			50	1.85 × 10^6^	100	5.4 × 10^−5^
HICC (Lyral^®^)	25	14.3	1.23 × 10^−2^			7.5	610	300	0.49
BPMP (Lilial^®^)	25	5.1	0.44 × 10^−2^			2.5	568	100	0.18
Alpha-isomethyl ionone	25	100	8.59 × 10^−2^			25	291	100	0.34
Formaldehyde	37.4	0.5	0.65 × 10^−3^	0.075	0.87 × 10^−2^				
Formaldehyde (synthetic urine)	2.75	0.5	0.70 × 10^−3^	0.075	0.94 × 10^−2^				

^1^ Extraction from shredded diapers with an organic solvent. Formaldehyde was also quantified in the extract of shredded diapers with synthetic urine. The value of 25 mg/kg for the fragrances corresponds to LOQ/2. ^2^ Limonene, human, in vivo [46]; benzyl salicylate, human skin, in vitro, [47]; HICC, human skin, in vitro, [48]; BPMP, human skin in vitro [49]; formaldehyde, monkeys, in vivo [43,44]. ^3^ Intake from diapers calculated according to ANSES extraction scenario 1 for an infant aged 0–6 months (body weight, 3.9 kg; 7.98 diapers/day; diaper weight, 24 g). ^4^ NOAELs or TDI of HICC, BPMP, alpha-isomethyl ionone and formaldehyde were adjusted for a fractional oral absorption of 50%. The fractional oral absorption of other compounds was assumed to be 100%. Abbreviations: HICC, hydroxyisohexyl cyclohexene carboxaldehyde; BPMP, butylphenyl methylpropional; TDI, tolerable daily intake; MOE, margin of exposure; NOAEL, no-observed-adverse-effect level.

## Data Availability

Concentrations of chemical substances in diapers used in this review can be found in the ANSES report (in French) [7].

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
