# Peer review of "Dermal Exposure to Hazardous Chemicals in Baby Diapers: A Re-Evaluation of the Quantitative Health Risk Assessment Conducted by The French Agency for Food, Environmental and Occupational Health and Safety (ANSES)"

_ijerph, 2022, doi:10.3390/ijerph19074159_

Round 1

Reviewer 1 Report

The manuscript entitled “Dermal exposure to hazardous chemicals in baby diapers: a critical review of the quantitative health risk assessment conducted by the French Agency for Food, Environmental and Occupational Health and Safety (ANSES)” presents a very detailed analysis of the risk assessment of chemicals potentially contained in baby diapers. The topic is extremely relevant and likely to be of interest for academic authors as well as regulatory bodies. Starting position for the review was a recent risk assessment of the French Agency for Food, Environmental and Occupational Health and Safety (ANSES).

Revision: The manuscript is very detailed and rich in information. The author should consider the implementation of some workflow graphs or schematic representations: this could maybe support readers who are less familiar with the concepts of regulatory risk assessment to follow the different lines of argument. This is especially true for the comparison of the different evaluation strategies that are at the core of the critical review and whose value goes beyond the application on a specific topic. In this light, major emphasis could be given, also in general terms, to the different strategies potentially applicable for the risk assessment with pros and cons of the different approaches. As obviously pointed out also by the author, different assumptions/workflows lead necessarily to different conclusions and have major impact on the final outcome. Such contextualized systematic analysis could also support in the future other risk assessors to identify critical steps which might potentially influence their conclusions and interpretations.

Author Response

I thank this reviewer for his very positive comments and his suggestion to include a figure or table summarizing and comparing the risk assessment conducted by ANSES and in this study. 

A Table (Table 1 in the revised manuscript) has been added in the Materials and Methods section to compare the risk assessment methodology used by ANSES with that adopted in this re-evaluation. This has been done separately for the PCDD/Fs and DL-PCBs, for PAHs and for other compounds detected or quantified in diapers except for the exposure data that this study took from the ANSES report.

Reviewer 2 Report

ijerph-1340624 – Dermal exposure to hazardous chemicals in baby diapers: a critical review of the quantitative health risk assessment conducted by the French Agency for Food, Environmental and Occupational Health and Safety (ANSES).

  • The manuscript has deficiency citation to similar works published before like:
  • Journal of Toxicology and Environmental Health, Part A. Safety evaluation of disposable baby diapers using principles of quantitative risk assessment. 72 (2009) 1262-271
  • Regulatory Toxicology and Pharmacolog. Exposure Factor considerations for safety evaluation of modern disposable diapers. 81 (2016) 183-193.
  • Regulatory Toxicology and Pharmacolog. Safety evaluation for ingredients used in baby care products: Consideration of diaper rash. 90 (2017) 214-221
  • Analyses of data/results and focus of aims should be further elevated.

I recommend this work for publication after minor revision.

Author Response

I thank the reviewer for his suggestions that help me to improve my manuscript.

As suggested, two new references have been added: Felter et al. 2017 and Rai et al. 2009. The reference of Dey  et al. 2016 was already included in the first version. 

As also suggested,  a Table (Table 1 in the revised manuscript) has been added in the Materials and Methods section to compare the risk assessment methodology used by ANSES with that adopted in this re-evaluation. This has been done separately for the PCDD/Fs and DL-PCBs, for PAHs and for the other compounds detected or quantified in diapers except for the exposure data that this study took from the ANSES report.

Reviewer 3 Report

This critical review is entirely based on the ANSES study solely, which I think is inappropriate. I think this is not a critical review, but a data reanalysis. If it is a data reanalysis, then the original ANSES authors should be listed as the author, not just cited. The author did not provide update information or novelty. In addition, the author should highlight the differences with the original report.

Author Response

As suggested by the reviewer, the expression "critical review" has been replaced in the title by "a re-evaluation" while in the abstract and throughout the manuscript it was replaced by "this study" or "this re-evaluation".  Regarding a possible co-authorship with ANSES, it is clear that the French agency will never sign a study contradicting its own conclusions that are based on the evaluation by a panel of independent experts, especially as these conclusions have received broad media coverage in France and beyond. 

Round 2

Reviewer 3 Report

This study by Alfred Bernard was to examine whether the exposure and risk assessment conducted by ANSES on risks related to the presence of hazardous chemicals in infant diapers contained potential flaws that could explain such a high exceedance of health reference values. Bernard re-evaluated the quantitative health risk assessment conducted by ANSES. He indicated that the quantitative health risk assessment conducted by ANSES was flawed by several scientifically unjustified risk assessment approaches and assumptions that had led to incredible risk estimates and to concentration limits in diapers hardly quantifiable by their analytical methods. The author also suggests there is a clear need to revisit the ANSES risk assessment by using more accurate exposure data, more toxicologically relevant endpoints and more realistic exposure scenarios.

This reanalysis is an important supplement and correction to the original research report, using a more reasonable and comprehensive method.

The manuscript was written and organized well. The revision is an improvement on the origin.